# MHC Class I Deficiency in Solid Tumors and Therapeutic Strategies to Overcome It

**DOI:** 10.3390/ijms22136741

**Published:** 2021-06-23

**Authors:** Elena Shklovskaya, Helen Rizos

**Affiliations:** Faculty of Medicine, Health and Human Sciences, Macquarie University, Sydney, NSW 2109, Australia; helen.rizos@mq.edu.au

**Keywords:** major histocompatibility complex (MHC), MHC-I, tumor antigens, tumor antigen presentation, immunotherapy, immune checkpoint blockade, NK cells, T-cell subsets

## Abstract

It is now well accepted that the immune system can control cancer growth. However, tumors escape immune-mediated control through multiple mechanisms and the downregulation or loss of major histocompatibility class (MHC)-I molecules is a common immune escape mechanism in many cancers. MHC-I molecules present antigenic peptides to cytotoxic T cells, and MHC-I loss can render tumor cells invisible to the immune system. In this review, we examine the dysregulation of MHC-I expression in cancer, explore the nature of MHC-I-bound antigenic peptides recognized by immune cells, and discuss therapeutic strategies that can be used to overcome MHC-I deficiency in solid tumors, with a focus on the role of natural killer (NK) cells and CD4 T cells.

## 1. Introduction

The immune system plays a critical role in preventing and controlling cancer growth. Anti-tumor immunity relies largely on CD8 T cell-mediated recognition of tumor antigens. Cancers have developed sophisticated strategies to escape immune-mediated control and this includes the downregulation or loss of antigens or the major histocompatibility class (MHC)-I molecules—the molecular structures presenting these antigens. Cancers can also promote an immunosuppressive microenvironment that disables T-cell killing via the expression of immune inhibitory molecules. Immunotherapy, also known as immune checkpoint blockade (ICB), helps the immune system regain control. Therapeutic antibodies targeting the immune inhibitory receptors Programmed Death-1 (PD-1) and Cytotoxic T Lymphocyte Antigen-4 (CTLA-4) induce durable responses in a proportion of patients with melanoma, lung, kidney, bladder and other cancers [1,2,3,4,5,6,7,8,9]. However, many patients progress while on treatment, and primary treatment resistance remains a major obstacle. Combined immunotherapy (anti-PD1 plus anti-CTLA-4) results in higher objective response rates [2] but is also associated with significant immune-related toxicities that are often severe and can be life-threatening [10,11].

Multiple biomarkers predictive of immunotherapy response and resistance have been proposed, including but not limited to the expression of MHC-I and MHC class II (MHC-II) on tumor cells [12,13,14,15], transcriptome and cell signatures of immune activation [16,17,18,19], tumor mutation burden [20,21,22,23], favorable gut flora [24,25,26,27] and combinations of these markers [28,29,30,31,32]. Our ability to predict immunotherapy response, however, remains disappointing [14,33] and there are currently no objective criteria to select patients who require the more efficient, but toxic combination immunotherapy. We have recently demonstrated that in advanced melanoma, patients with low expression of MHC-I on the surface of tumor cells, were unlikely to benefit from anti-PD-1 monotherapy but could still respond to the combined treatment (anti-PD-1 and anti-CTLA-4) [34]. It is unclear how the combined immunotherapy allows the patient’s immune system to overcome MHC-I downregulation or loss. In this review, we examine the mechanisms controlling MHC-I expression and antigen presentation in tumor cells and explore therapeutic strategies that may allow the immune system to overcome MHC-I loss in tumor cells.

## 2. Cancer Immune Escape via MHC-I Downregulation

### 2.1. Overview of the MHC-I Expression

MHC-I molecules are expressed on the surface of all nucleated cells where they form trimeric complexes consisting of the membrane-linked MHC-I heavy chain, a soluble invariant light chain β2-microglobilin (B2M) and a short peptide antigen bound in a groove formed by the membrane-distal (α1 and α2) domains of the MHC-I heavy chain. Peptide/MHC-I (pMHC-I) complexes are recognized by CD8 T cells. Complexes containing altered (“foreign”) peptides activate CD8 T cells to kill the cellular targets displaying such aberrant complexes. In contrast, unaltered self-peptide/MHC-I complexes provide tonic signals required for the survival of naïve CD8 T cells [35]. In cancer, cellular stress or mutations accumulated during oncogenesis, provide a steady source for novel peptides that are loaded onto MHC-I molecules, allowing the immune system to recognize and eliminate malignant cells. Categories of tumor antigens and their generation are discussed in Section 3.

The heavy chains of MHC-I, also known as HLA in humans (HLA stands for Human Leukocyte Antigen) are encoded by classical *MHC-I* genes, *HLA-A*, *HLA-B* and *HLA-C*. Classical *MHC-I* genes are highly polymorphic, with over 20,500 alleles reported in the IPD-IMGT/HLA Database [36] (website: http://hla.alleles.org/nomenclature/stats.html, accessed on 5 May 2021). The non-classical *MHC-I* genes encoding HLA-E, -F and -G are less polymorphic, being encoded by <400 alleles. Allelic variants of the MHC-I molecules can differ in as many as 20 amino acids. These differences mainly map to within and around the peptide-binding grooves (α1 and α2 domains) resulting in distinct repertoires of bound peptide ligands forming unique “MHC-I ligandomes” [37].

pMHC-I complexes are assembled in the endoplasmic reticulum (ER) via a complex pathway (Figure 1). Peptides are typically generated in the cytosol by the proteasome and transported to the ER by the Transporter associated with Antigen Processing (TAP). Peptides are trimmed to the optimal length of 8–10 (usually 9) amino acids by the ER aminopeptidases (ERAP)1 and ERAP2 and loaded onto the MHC-I molecules with the help of the peptide-loading complex comprising TAP subunits TAP1 and TAP2, tapasin, ERp57 and calreticulin. Finally, peptide-loaded MHC-I complexes are delivered to the plasma membrane where they are displayed to CD8 T cells. Surface pMHC-I complexes are subject to internalization and endosomal recycling. Deficiency in either *MHC-I* heavy chain, *B2M* or any of the subunits of the peptide-loading complex will affect pMHC-I complex display on the cell surface, potentially avoiding recognition by CD8 T cells [38]. Further details on MHC-I assembly and peptide loading are provided in [39,40].

Expression of MHC-I molecules is controlled at multiple levels. The master regulator of the *MHC-I* genes, Nucleotide-binding oligomerization domain-Like Receptor family Caspase recruitment domain containing 5 (NLRC5), also known as the Class I Transactivator (CITA), controls both baseline and interferon (IFN)γ-induced expression of MHC-I molecules [41]. Similar to the Class II Transactivator (CIITA), another member of the NLR family that regulates expression of the MHC-II genes, NLRC5 does not bind DNA directly but assembles a complex known as the “CITA enhanceosome” to induce *MHC-I* expression (Figure 1) [42]. NLRC5 associates with chromatin remodelling enzymes, such as histone acetyltransferases 2A and 2B that regulate *MHC-I* expression at the epigenetic level. Other epigenetic mechanisms suppress *MHC-I* expression in cancer, such as DNA hypermethylation at the promoters of *MHC-I* heavy chains/antigen-presenting machinery (APM) genes, and trimethylation of lysine 27 on histone 3 (H3K27me3 repressive mark) by the Polycomb Repressive Complex 2 (PRC2) [43,44].

Interferons (IFN), in particular IFNγ, increase expression of pMHC-I complexes on tumor cells by concomitantly upregulating transcription of *MHC-I* heavy chains, *B2M*, most components of the peptide-loading complex and three unique components of the immunoproteasome (low-molecular-weight polypeptide (LMP)2/Proteasome 20S subunit beta (*PSMB*)*9*, LMP7/*PSMB8* and the multicatalytic endopeptidase complex-like (MECL)-1/*PSMB10*). These unique immunoproteasome components replace the standard β1, β2 and β5 proteasome subunits to enhance the generation of MHC-I-compatible peptides [45]. This effect is mediated by several transcription factors, including the p50 and p65 subunits of the nuclear factor kappa-light-chain-enhancer of activated B cells (NK-kB) and interferon regulatory factor 1 (IRF1) that bind to the Enhancer A and the interferon-stimulated response element (ISRE), respectively, in the *MHC-I* promoter to stimulate the expression of *MHC-I* heavy chains and the APM genes (Figure 1) [40,46]. In addition, IFNγ stimulates expression of *NLRC5* that assembles the CITA enhanceosome with transcription factors binding to W/S, X1, X2 and Y1 boxes in the *MHC-I* promoter (Figure 1) [47].

### 2.2. MHC-I Expression in Cancer: Implications for Immunotherapy

*MHC-I expression is commonly altered in cancer*. Downregulation or loss of classical MHC-I molecules is frequently observed and is considered the most common mechanism of tumor immune escape, as it leads to poor tumor recognition and limited killing by cytotoxic T cells. In contrast, tumors infiltrated by CD8 T cells often demonstrate higher MHC-I expression compared to the adjacent tissue, an increase that has been linked mechanistically to IFNγ secreted by activated T cells in the tumor microenvironment [40,48]. While the significance of loss or downregulation of MHC-I expression in tumors has long been recognized [48,49,50,51], the precise molecular mechanisms underlying this loss have only recently been fully explored. The main challenge in recognizing allele-specific MHC-I alterations in tumors lies in the polymorphic nature of the MHC-I locus. Not until the development of computational zygosity prediction algorithms, such as Polysolver [52], LOHHLA [53] and somatic-germline-zygosity (SGZ) algorithm [54], has the analysis of copy number alterations and mutations in the polymorphic *HLA* genes become possible.

*Germline MHC-I diversity*. Maximizing tumor antigen recognition by the immune system favors better patient outcomes after immunotherapy. MHC-I expression is co-dominant, thus up to six *MHC-I* alleles can be expressed in any individual heterozygous for *HLA-A*, *-B* and *-C*. Studying *MHC-I* diversity in >1500 patients (mostly with melanoma and lung cancer) treated with immune checkpoint inhibitors, Chan and colleagues demonstrated that homozygosity at one *MHC-I* locus (either *HLA-A*, *-B* or -*C*) was associated with a significant reduction in overall survival, compared with the “full house” *MHC-I* [55]. Although some *MHC-I* “supertypes” were associated with either better (*HLA-B44*) or worse (*HLA-B62*) outcomes in this study [55], in general patients with high *MHC-I* diversity had better immunotherapy outcomes than patients with low *MHC-I* diversity, and patients with complete *MHC-I* heterozygosity and evolutionary diverse (i.e., most dissimilar) *MHC-I* alleles had the best outcomes [56]. Furthermore, patients homozygous at one *MHC-I* locus who also had the least mutated tumors, had the worst immunotherapy outcomes [55]. Thus, the more diverse MHC-I repertoire, the more different peptides MHC-I molecules can bind, ultimately resulting in superior recognition by CD8 T cells and higher clonality of tumor-reactive CD8 T cells. Importantly, the effect of germline HLA zygosity on immunotherapy outcomes may be tumor-specific, as no such effect was observed in a recent study of 240 immunotherapy-naïve lung cancer patients treated with immune checkpoint inhibitors [54].

*Somatic mutations in the MHC-I genes*. Next to *HLA-I* LOH, mutations in *MHC-I* genes are commonly observed in cancer. A comprehensive analysis of samples from The Cancer Genome Atlas project (TCGA) demonstrated that *MHC-I* mutations were present in 3.3% (266/7930) of studied cases [52]. The highest frequency of *MHC-I* mutations was found in head and neck- and lung squamous carcinomas (10% and 6.9%, respectively), while the lowest was in glioblastoma, ovarian and breast cancer (0–0.6%) [52]. The majority of mutations (40%) were mapped to the region encoding the α3 domain of the MHC-I molecule that interacts with the CD8 co-receptor on the T cell, while 35% of mutations were mapped to the regions encoding the α1 and α2 domains, specifically the peptide-binding pockets [52]. These mutations would therefore result in either (i) complete abrogation of peptide loading and thus cell surface pMHC-I display, or (ii) change in the repertoire of peptides bound to the mutated MHC-I molecule (Figure 1g). Mutations in *MHC-I* result in loss of CD8 T-cell binding or loss of clonal antigen recognition. There are important challenges in recognizing allele-specific *MHC-I* mutations in tumors, such as their focal (rather than universal) distribution, varying subclonal frequencies, the polymorphic nature of the *MHC* locus and the frequently unaltered gene expression [52,53]. Interestingly, subclonal *MHC-I* mutations and *MHC-I* LOH were both associated with high CD8 (cytolytic) activity in the affected lesions [52,53], indicative of the ongoing tumor immune editing that drives subclonal evolution, particularly in the context of immunotherapy. Importantly, a subclonal distribution of *MHC-I* mutations and a high degree of tumor heterogeneity may explain a lack of correlation between the occurrence of *MHC-I* mutations and immunotherapy response in melanoma [14], while rare amplifications in *MHC-I* genes were associated with good immunotherapy response [14], indicative of the importance of high MHC-I expression for immune-mediated tumor control.

*Downregulation or loss of MHC-I expression due to genetic events*. Assembly of all pMHC-I complexes is critically dependent on the availability of the invariant chain, B2M. Consequently, LOH or mutations in the *B2M* gene result in the downregulation or loss of multiple MHC-I molecules. The *B2M* gene is commonly mutated across multiple cancers with the highest mutation rate reported in uterine, stomach, colorectal and breast cancer [16]. In addition, *B2M* mutations were strongly associated with patient progression on immunotherapy in melanoma [57,58,59,60], colorectal [59,61,62] and bladder cancer [59]. A complete loss of tumor MHC-I expression is typically the result of two genetic events, a mutation in one copy of the *B2M* gene combined with the loss of the second copy (*B2M*-LOH) [59]. As the *B2M* gene is located on chromosome 15 (15q21) and *MHC-I* on chromosome 6 (6p21), the genetic alterations in the *B2M* and *MHC-I* loci are unrelated. A complete loss of *B2M* conferred immunotherapy resistance in some [60,62,63] but not in other studies [14]. On the whole, while the loss of one copy of *B2M* is relatively common across multiple cancers [16], a complete loss of *B2M* expression is uncommon [16,60,63,64,65], most likely because a complete loss of MHC-I makes the tumor susceptible to NK cell-mediated killing (more details in Section 4.2.1). Significantly, T-cell activity resulting from pMHC-I recognition appears to be the main driver of genetic events resulting in tumor MHC-I downregulation or loss (mutations or LOH in *B2M*, *MHC-I* and APM genes) as these were significantly correlated with high T-cell activity [16,60] and were initially subclonal [on repeat biopsies], consistent with the emergence of resistant clones via immune editing within the otherwise immune-competent environment [16,60,66,67].

*Downregulation or loss of MHC-I expression due to loss of response to IFN**γ*. High expression of MHC-I by tumor cells is frequently observed in immunologically active cancers [13,15,68,69,70]. High tumor MHC-I expression was strongly correlated with CD8 T-cell infiltration and activity, and with expression of IFNγ-upregulated genes (“IFNγ signatures”) in multiple studies [16,17,19,67,69,71,72,73,74]. Upregulation of IFNγ signature early during therapy correlated with improved outcomes of immune checkpoint blockade in melanoma [75]. Mutations in the IFNγ receptor *IF**NG**R1* [76], signaling adaptors *JAK1* and *JAK2* [63,65,77,78] or components of the IFNγ signaling cascade, such as *STAT1* and *IRF1* [79,80] were associated with cancer progression and immunotherapy resistance. Yet in melanoma and lung cancer—the two cancers where cancer evolution during immunotherapy is studied best—genetic alterations in IFNγ signaling pathways were relatively uncommon [60,63,64,65] and some of the mutations (such as in *STAT1*) occurred in both responders and non-responders [60,81,82], indicating that loss of INFγ responsiveness does not necessarily favor tumor progression.

*Alterations in NLRC5 expression in cancer*. IFNγ stimulates MHC-I expression in cancer through multiple ways, of which the induction of NLRC5 expression followed by the formation of CITA enhanceosome (Figure 1) is the principal mechanism of IFNγ-induced increase in MHC-I (reviewed in [83]). Four independently generated *Nlcr5*-deficient mouse strains demonstrated the principal role of NLRC5 in the regulation of both classical and non-classical *MHC-I* genes and APM components [84,85,86,87]. Interestingly, constitutive MHC-I expression was also affected in *Nlcr5*-deficient mice [47,87,88], indicating that NLRC5 maintains baseline as well as IFNγ-induced MHC-I expression. CD8 T cell-mediated killing of *NLRC5*-deficient targets was dramatically reduced compared with NLRC5-sufficient cells [85]. *NLRC5* gene expression strongly correlates with expression of *MHC-I* and APM components across multiple cancers including prostate, lung, melanoma, thyroid, breast, uterine and ovarian cancer [89,90]. Genetic alterations of *NLRC5* (copy number loss, somatic mutations and promoter methylation) dramatically reduced MHC-I expression across 16 cancer types [89]. Furthermore, reduced expression of *NLRC5* correlated with poor survival in melanoma and bladder cancer patients, while an inverse correlation was observed in glioma [89]. Paradoxically, high *NLRC5* expression appeared to favor cancer progression in hepatocellular carcinoma, by stimulating tumor proliferation via the AKT pathway and promoting epithelial-to-mesenchymal transition (EMT) via Wnt/β-catenin signaling [91,92]. The relevance of these findings in a broader context of EMT in cancer is unclear, although we and others have demonstrated a mechanistic link between EMT and impaired MHC-I expression or induction in melanoma [93], prostate cancer [94] and a mouse model of mammary carcinoma [95], through the upregulation of the transcription factor Snail driving activation of transforming growth factor (TGF)β and suppression of NF-kB signaling [93,94]. Overexpression of *NLRC5* in tumor cells can dramatically enhance their immunogenicity, such as overexpression of *Nlrc5* in a mouse model of melanoma resulted in improved tumor clearance [78]. Finally, radiation appears to be able to induce MHC-I expression by directly upregulating *NLRC5*, as well as by activating interferon signaling [96].

*Transcriptional and post-transcriptional downregulation of MHC-I expression*. Transcriptional downregulation of *MHC-I* expression is commonly observed in cancer and it is accompanied by reduced expression of the APM components, as the two are strongly correlated [15,97,98]. MHC-I downregulation in cancer is driven by several mechanisms, including (i) reduced IFNγ production by tumor-specific T cells due to tumor antigen loss (such as the result of de-differentiation) and/or T-cell functional inactivation through the expression of multiple inhibitory checkpoints [58,99], (ii) alterations in IFNγ signaling in cancer cells, including both loss of responsiveness via inactivation of IFNγ signaling pathways and sustained intrinsic IFNγ signaling [46,74,76,99]; (iii) alterations in NLRC5 expression in cancer cells [83]; (iv) epigenetic regulation of *MHC-I* expression, through DNA hypermethylation, histone deacetylation and trimethylation of H3K27 [43,44,100] (Figure 1). These mechanisms often overlap in emerging tumor subclones, driving intra- and inter-lesional heterogeneity, clonal evolution and treatment resistance [93,101,102]. MHC-I expression is also subject to posttranscriptional, allele-specific modifications. For example, expression of *HLA-A2* is regulated by mRNA-binding proteins MEX3B [103] and MEX3C [104] that bind to the 3′ untranslated region (3′-UTR) and destabilize *HLA-A2* mRNA [103,104]. It is not yet clear whether similar mechanisms control expression of other HLA alleles.

Epigenetic mechanisms of tumor MHC-I downregulation have been studied in detail as they can potentially be targeted therapeutically. Hypermethylation of the promoters of *MHC-I* heavy chains, *B2M*, APM components and/or *NLRC5*, leads to the suppression of *MHC-I* expression that is can be overcome with DNA methyltransferase inhibitors (DNMTi) as demonstrated for breast, lung, colon, thyroid cancers, human papillomavirus (HPV)-related cancers, sarcomas and gliomas [105,106,107,108,109]. *MHC-I* epigenetic silencing due to histone deacetylation can be reversed with histone deacetylase inhibitors (HDACi), as demonstrated for melanoma and glioma [110,111]. Finally, deposition of the H3K27me3 repressive mark by the polycomb repressor complex 2 (PRC2) subunit Enhancer of zeste homologue 2 (EZH2) suppresses both basal and IFNγ-induced MHC-I expression [43,44]. *EZH2* is recurrently mutated in some cancers and highly expressed in others; activating mutations in the *EZH2* gene promote cancer growth and progression through several mechanisms, including MHC-I downregulation [112]. *EZH2* knockdown in mouse models of cancer or targeting EZH2 with specific inhibitors allowed restoration of MHC-I expression in melanoma, B cell lymphoma and lung cancer [43,44,112,113]. Remarkably, some tumors that did not exhibit alterations in *EZH2* (i.e., gene amplification/activating mutation) remained sensitive to EZH2 inhibition revealing an alternate regulatory mechanism; a long non-coding (lnc)RNA *EPIC1* was found to bind EZH2 leading to the epigenetic silencing of *IFNGR1*, *TAP1/2*, *ERAP* and *MHC-I* genes [114].

Finally, pMHC-I downregulation on tumor cell surface can occur via alterations in intracellular protein trafficking. It is estimated that pMHC-I complexes remain on the cell surface for about 7–12 h [115,116], with longer cell surface residency directly correlated to the complex immunogenicity [115]. Oncogenic mutation *BRAF^V600E^* is found in up to 50% of human melanomas; mutant BRAF protein induces internalization of pMHC-I complexes and their intracellular sequestration within endo-lysosomal compartments, facilitating tumor immune evasion via reduced surface pMHC-I display [117]. Accordingly, treatment with a BRAF inhibitor rapidly restored MHC-I expression and improved T-cell recognition [117]. Similarly, transmembrane protein Myelin and lymphocyte 2 (MAL2) that is frequently overexpressed in breast cancer [118] associates with pMHC-I complexes to promote endocytosis and degradation [119]; *MAL2* knockdown improved tumor recognition and T-cell activation [119].

## 3. Tumor Antigens Recognized by the Immune System

T cells recognize a diverse repertoire of antigens bound to MHC-I molecules. This repertoire is determined not only by the diversity of MHC-I alleles expressed on the surface of tumor cells, but also by genomic instability and alterations in peptide processing that are a direct result of carcinogenesis and tumor immune evolution. Tumor antigens that are recognized by the immune system, can be classified into two large groups: tumor-associated antigens and tumor-specific antigens. Tumor-associated antigens are expressed in the tumor, but they can also be expressed in healthy tissues. Tumor-specific antigens are not expressed in healthy tissues. An overview of the main subtypes of tumor antigens and methods of their identification is provided in Table 1.

TAAs are essentially self-proteins that are recognized by the immune system because of their unusually high or atypical expression. TAAs fall into four groups, (i) overexpressed proteins, (ii) tissue differentiation antigens, (iii) germline antigens and (iv) epitopes associated with impaired peptide processing. Since TAAs are shared between the tumor and healthy tissues, therapeutic targeting of TAAs is associated with a potential off-tumor, on-target toxicity.

### 3.1. Tumor-Associated Antigens (TAAs)

*Overexpressed antigens*. Genes that are essential for tumor survival, are often overexpressed in tumors and their products can trigger immune responses. Examples include human epidermal growth factor receptors *EGFR* (*HER1*) and *ERBB2* (*HER2*) whose overexpression in head and neck, ovarian, cervical, bladder and esophageal cancer (*EGFR*), or breast and ovarian cancer (*HER2*) is associated with poor prognosis [145,146,147]. Carcinoembryonic antigen is expressed in several epithelial cancers [148,149]. Overexpressed TAAs represent an attractive therapeutic target because they are required for tumor cell survival and therefore tend to be retained in cancer cells under immune selection pressure.

*Tissue differentiation antigens* are TAAs whose expression is restricted to tumors and their tissue of origin. The first tissue differentiation antigens identified, MART-1 and gp100, belong to the melanocytic lineage and are expressed in melanoma but also normal melanocytes in the skin, eye and inner ear [150,151]. Therapeutic targeting of tissue differentiation antigens poses the risk of toxicities to normal tissues. For example, vitiligo, ocular toxicity and hearing loss due to destruction of normal melanocytes were observed in some melanoma patients after adoptive transfer of ex vivo expanded T-cell products recognizing melanocytic antigens [152,153,154,155,156], in particular after adoptive transfer of genetically engineered T cells with high affinity for MART-1 or gp100 [121]. Some differentiation antigens are overexpressed in particular biological contexts, such as breast tissue antigen Ankyrin repeat domain 30A (also known as NY-BR-1) whose expression is strongly correlated with estrogen-receptor status and is often low in triple-negative breast cancer [157,158,159].

*Cancer germline antigens*, also known as cancer-testis (CT) antigens, are the products of genes normally expressed by gametes and trophoblasts only; these genes are epigenetically silenced in adult tissues but are aberrantly expressed in a range of cancers (reviewed in [160]). The first CT antigen was discovered in 1991 in a patient with melanoma and termed MAGE-A1 [161]; the current list includes 277 CT antigens (CT database: http://www.cta.lncc.br, accessed on 5 May 2021). The CT antigen New York Esophageal Squamous Cell Carcinoma-1 (NY-ESO-1) is one of the most immunogenic CT antigens and has been used extensively in cancer vaccine development. NY-ESO-1 is expressed by a large fraction of neuroblastomas (82%), synovial cell sarcomas (80%), metastatic melanomas (46%) and ovarian cancers (43%), as well as a proportion of other cancers (bladder, esophageal, hepatocellular, head and neck, ovarian, prostate and breast cancers) [162]. CT antigen-based therapeutic vaccines could generate high frequencies of reactive T cells in the blood but were generally ineffective in achieving objective responses; in contrast, adoptive therapies with genetically engineered T cells targeting NY-ESO-1, led to objective responses in a large proportion of patients with melanoma and synovial cell carcinoma, with no off-target toxicities [121]. However, the responses were mostly transient, likely owing to incomplete tumor antigen expression particularly in metastatic disease [121].

*T-cell epitopes associated with impaired peptide processing*. In immune-edited tumors with MHC-I downregulation and/or *TAP* deficiency, T-cell epitopes associated with impaired peptide processing (TIEPP) contribute to the generation of an alternative tumor antigen repertoire [124]. TIEPP are non-mutated antigens generated via an alternative processing route, best studied for the processing of signal peptides [163,164]. After protein integration into the ER membrane, a fragment of the signal peptide is liberated by the enzyme signal peptide peptidase, uploaded onto MHC-I and presented as a pMHC-I complex. Immune responses to TIEPP antigens have been documented in humans and mice [165,166]. While spontaneous priming of TIEPP-specific T cells may never occur in vivo [due to low pMHC-I expression levels], T cells activated via therapeutic intervention could control tumor growth in MHC-I/MHC-II humanized mouse models [126,167], while transient tumor-targeted silencing of TAP allowed immune cell activation and improved tumor control in vivo [168].

### 3.2. Tumor-Specific Antigens (TSAs)

This group includes tumor antigens that are recognized by the immune system as “foreign”, as their peptide sequence differs from self-proteins. TSAs include (i) endogenous retroviruses, (ii) oncogenic viruses and (iii) neoantigens.

*Endogenous retroviruses* (ERVs) occupy a large fraction (estimated 8%) of the mammalian genomes and can be expressed in tumors due to oncogenic activation and epigenetic de-repression. Human ERVs (hERVs) are relics of ancient retroviral infections that integrated into the human genome throughout evolution. Over 3100 *hERVs* are thought to be transcriptionally active across multiple cancers [169,170]. hERVs can induce immune responses to tumors by activating innate viral sensing mechanisms, while also generating peptides for T-cell recognition. Tumor *hERV* expression is associated with T-cell infiltration and high cytolytic activity [16]. *hERVs* are highly expressed in cancers of viral origin, such as a subset of cervical cancers (>90%) and a subset of head and neck cancers (12%) due to human papilloma virus infection, hepatitis B virus-linked liver cancers (25%) and Epstein–Barr virus-linked stomach cancers (8%) [16]. In addition, kidney cancer (clear cell renal cell carcinoma, ccRCC) express multiple *hERVs*. Cancer *ERV* expression, confirmed through ribosome profiling combined with validation for MHC binding and identification of ERV epitope-specific T cells, correlated with anti-PD-1 immunotherapy response in ccRCC [170], which is remarkably high in this cancer despite its relatively low tumor mutation burden (TMB) [22]. Adoptive cell therapy with T cells genetically engineered to recognize an hERV variant highly and specifically expressed in ccRCC, is currently being tested in phase I trials (clinicaltrials.gov (accessed on 5 May 2021) identifier: NCT03354390). Another example of the TMB^low^ cancer that responds well to immunotherapy due to tumor *hERV* expression, is Merkel cell carcinoma, a tumor linked to Merkel cell polyomavirus [22].

*Oncogenic viruses* are associated with a number of solid cancers including hepatocellular carcinoma (hepatitis B and hepatitis C viruses), Kaposi sarcoma (human herpes virus 8), oropharyngeal and cervical cancers (HPV) [128]. Oncoviral antigens are not expressed in normal tissues and are therefore highly tumor-specific, as well as shared between patients making them attractive vaccine candidates. Therapeutic targeting of HPV16 with a DNA-based vaccine generated durable responses in a subset of patients with HPV16-positive squamous cell carcinoma of the head and neck [171] and high grade vulvar and vaginal neoplasia [172].

*Neoantigens* represent the most studied and the most clinically significant group of TSAs to date. Neoantigens are derived from non-synonymous mutations, insertions and deletions (indels) that lead to frameshifts, and structural variants (reviewed in [173]). Neoantigens are essentially the products of mutations that accumulate in the DNA of cancer cells during tumor evolution. Neoantigens are not expressed in healthy tissues, nor are they present in the thymus during T-cell development, so that T cells with reactivity for neoantigens are not subject to central tolerance. For these reasons, neoantigens can generate potent anti-tumor immune responses that are both individualized and highly tumor-specific [173].

Neoantigens can induce both CD4 and CD8 T-cell responses, and reactivity of one or both T-cell subsets against mutated tumor peptides was confirmed initially by testing tumor-infiltrating T-cell (TIL) products expanded from melanoma [174,175,176], epithelial cancer [177], cervical cancer [178] and colorectal cancer biopsies [179,180]. T-cell reactivity against tumor neoantigens was subsequently demonstrated to underlie clinical responses to immunotherapies, such as CTLA-4 blockade in melanoma [181] and PD-1 blockade in non-small cell lung cancer [182]. Multiple studies used a molecularly defined neoantigen load that was correlated with immunotherapy response in multiple tumor types [16,181,182,183,184,185]. Finally, vaccination with individualized neoantigen vaccines produced objective responses in melanoma [185,186].

While neoantigen load alone was associated with but not predictive of patient outcomes in anti-CTLA-4 treated melanoma [181,183] and anti-PD-1 treated lung cancer cohorts [182], neoantigen load corrected for the predicted affinity of mutant peptides to bind MHC-I (termed differential agretopicity), better-predicted immunotherapy responses in the same patient cohorts [187]. High neoantigen load was generally correlated with high TMB, providing a rationale for using TMB as a predictor of immunotherapy response across different cancers [22,183]. Indeed, a global analysis of the genome of over 17,000 cancers demonstrated that the most mutated cancers (>10 mutations per megabase (Mb)) were also the most immunogenic ones [188]. A correlation between TMB and immunotherapy response in multiple studies [20,21,22,181,182,183,189,190,191,192] has led to the development of methods and computational tools to allow rapid TMB assessment [33]. TMB is currently calculated as the number of non-synonymous [germline excluded] coding mutations per exome (~38 Mb) using whole-exome sequencing (WES) but can also be also be inferred from RNA sequencing next-generation sequencing (RNAseq-NGS) expression data (Foundation Medicine: synonymous and non-synonymous coding mutations and short indels in intronic regions, 234 genes/~1.1 Mb; MSK-IMPACT: somatic exonic mutations and oncogenic drivers, 468 genes/~1.2 Mb) [23,33]. There is currently no standard TMB cutoff definition for patient stratification; FoundationOne defines ≥20 mutations/Mb as TMB-high, and ≤5 mutations/Mb as a TMB-low cutoff [33].

The rationale for using TMB as a predictor of immunotherapy response is obvious. Neoantigens are produced as a result of non-synonymous mutations, tumors with a greater mutation number (that is, high TMB) have a higher likelihood of producing neoantigens that are presented as pMHC complexes on the surface of tumor cells and trigger T-cell responses that may result in tumor eradication [16,23]. However, while there is a general correlation between high TMB and immunotherapy response [33], TMB predictive value in individual patients is limited, as only half of patients with TMB^high^ tumors respond to immunotherapy [29] while responses are also observed in patients with TMB^low^ tumors [22,170]. Several factors may be responsible for poor predictability of immunotherapy efficacy, which is ultimately determined by T-cell reactivity against tumor neoantigens. The first is antigen clonality. Clonal neoantigens were better than subclonal neoantigens at eliciting T-cell response in NSCLC and melanoma, and recognition of clonal antigens was associated with durable clinical benefit following checkpoint blockade therapy [193]. Second, neoantigens differ dramatically in their ability to trigger anti-tumor T-cell responses. For example, somatic mutations at hot spot positions in cancer driver genes —which are relatively common in tumors—are associated with low MHC binding ability; such mutations tend to occur in protein regions containing amino acid residues with weak MHC binding properties (*KRAS G12C/D/V* and *G13D*, *TP53 R175H* and *H179R*), or the driver substitution itself lowers MHC binding (*BRAFV600E*, *TP53 Y220C*) [194]. Currently, there are no computational tools that allow robust prediction of neoantigen immunogenicity, and different epitope prediction pipelines yield different outputs [173]. Third, recognition of neoantigens requires MHC expression. Low MHC expression, and/or loss of allelic expression (Section 2.2) will limit pMHC-I expression on the surface of tumor cells, with the resulting lack of T-cell recognition. Fourth, T-cell exclusion such as seen in cold tumors, will prevent pMHC-I recognition by T cells. Fifth, T-cell clonal competition, expression of multiple immune inhibitory molecules on antigen-specific T cells and/or immunosuppressive microenvironment variably restrict T-cell activation after pMHC-I recognition by the cognate T-cell receptor. Sixth, some neoantigens may promote T-cell tolerance instead of activation, although this is more likely to happen for MHC-II binding peptides. Finally, intratumoral heterogeneity, including genetic, epigenetic and microenvironment heterogeneity may influence response and the predictive value of TMB.

Apart from the functional assays based on T-cell activation—the ultimate measure of neoantigen immunogenicity, detection of neoantigens in the HLA ligandome (that is, detection of peptides bound to cell surface HLA molecules as confirmed by proteomic analysis) is the next best measure of neoantigen’s relevance for immune recognition [195]. Currently available neoantigen prediction pipelines, based on the analysis of exonic mutations (Table 1), identify only a small fraction (1–3%) of neoantigens in the HLA ligandome. This raises an important question: Are we looking in the right places?

### 3.3. Alternative TSAs

In addition to classical neoantigens derived from the gene coding sequences, a large class of alternatively expressed TSAs (aeTSAs) is emerging. The Encyclopedia of DNA Elements (ENCODE) project has determined that about three-quarters of the human genome is capable of being transcribed, with a range of expression spanning five to six orders of magnitude [196]. Some of these transcription events give rise to aeTSAs. Alternative TSAs include antigens derived from the apparently noncoding regions, mutational frameshifts, alternative splicing and/or editing of RNA transcripts, non-canonical translation and peptide slicing [140,141,142,196]. Alternative TSAs are less well defined than classical neoantigens, but accumulating evidence suggests that they play a major role in shaping anti-tumor immune responses, while they are largely ignored by currently accepted practices of estimating tumor immunogenicity.

Noncoding regions are the major source of targetable TSAs. In a recent study, Laumont et al. used a proteogenomic approach to screen mouse and human cancer cell lines for potential TSAs [142] (Table 1). The approach combined tandem mass spectrometry with tumor-specific RNA sequencing and screening against the medullary thymic epithelial cell (mTEC) library [to filter out promiscuous transcripts that drive thymic negative selection and central tolerance] [197]. Remarkably, 90% of the identified TSAs were derived from the “noncoding” regions such as RNAs containing single base mutations, intergenic deletions and endogenous retroelements, the latest representing “public” epitopes potentially shared between tumors. Importantly, immunization with 3 individual aeTSAs conferred protection in a mouse model of transplantable lymphoma, with the endogenous retroelement-derived TSA producing the best anti-tumor immunity [142]. Yewdell and colleagues used a different proteogenomic approach combining mass spectrometry, ribosomal sequencing [of ribosome-protected RNA fragments] and RNA sequencing, to determine the contribution of non-canonical translation to the generation of pMHC-I complexes in human B cell lymphoma [141]. Around 17% of proteins identified were non-canonical proteins, of which the majority (>70%) were cryptic proteins, derived from transcripts presumed to be non-coding (pseudogenes, non-coding RNAs, processed transcripts, intergenic regions and introns) [141]. Importantly, cryptic proteins were unstable (hampering their detection by traditional methods) yet generated MHC-I peptides much more efficiently than canonical proteins, demonstrating preferential access to the MHC-I loading machinery, possibly due to their susceptibility to proteasomal degradation (Figure 1) [141]. Once loaded onto MHC-I molecules, the peptides were protected from proteolytic degradation resulting in a greatly improved half-life [141] and potentially, immunogenicity.

In addition to aeTSAs described above, there is likely to be an additional pool of intron-encoded proteins translated in the nucleus [198]. Finally, apart from aeTSAs generated from non-coding regions and transcription products, aeTSAs can be generated post-translationally by proteasome-catalyzed peptide splicing. Such peptides are derived from hot spots within antigens and in a recent study of the MHC-I immunopeptidome of colon and breast cancers, they represented almost 20% of all peptides detected [199].

Neoantigens are increasingly being considered part of complex therapeutic strategies, such as in tumor vaccines. Neoantigen prediction tools and approaches are therefore being continuously improved (reviewed in [200]). The peptidome data, combined with transcript abundance and peptide processing data are increasingly used to improve the available tools used for the prediction of peptides binding a given HLA allele (Table 1). Furthermore, there is a concerted effort to standardize neoantigen prediction algorithms to increase reproducibility [173]. Consortia such as TESLA (Tumor Neoantigen Selection Alliance) with over 30 participating groups from academia, nonprofit institutions and industry, work on developing the best approaches for the identification and characterization of immunogenic neoantigens [173]. Inclusion of aeTSAs as part of neoantigen-based therapeutic strategies will broaden the reach of immunotherapies to potentially less immunogenic cancers, while the discovery of common neoantigens (including eaTSAs) may allow the development of off-the-shelf therapeutic products.

## 4. Therapeutic Strategies Directed at Overcoming Tumor MHC-I Downregulation or Loss

Downregulation or loss of MHC-I expression is a well-recognized mechanism of tumor immune escape. Partial loss of MHC-I reflects tumor immune evolution and is frequently observed in advanced cancer, such as in metastatic lesions in both immunotherapy-naïve patients and those who progressed on immunotherapy [13,14,15,34,50,63,93,192,201]. Factors that regulate baseline and IFNγ-induced MHC-I expression on tumor cells are consistently identified on high-throughput assays (such as in vitro and in vivo CRISPR screens) as important mediators of tumor sensitivity to immune-mediated killing, while loss of these factors mediates immunotherapy resistance [65,103,202,203,204]. Although advanced tumors may lose sensitivity to T-cell mediated killing through other mechanisms, such as loss of TAAs and TSAs as a result of dedifferentiation, this also creates new therapeutic vulnerabilities [205], including the generation of novel TSAs that may be recognized by the immune system. In this case, robust tumor MHC-I expression is still required to achieve immune recognition. Could MHC-I downregulation or loss be overcome therapeutically?

Garrido and colleagues have proposed that tumor MHC-I downregulation or loss is categorized as “soft” and “hard” alterations, depending on whether the loss of MHC-I expression is reversible (such as transcriptional downregulation) or not (for instance, loss of both copies of a gene such as *B2M*) [50,206]. We will therefore divide the strategies of dealing with MHC-I loss into two large groups: (i) restoration of pMHC-I cell surface expression and CD8 T-cell reactivity (for “soft” lesions), and (ii) re-directed immune recognition (for “hard” lesions). In the following section, we review the strategies that can be used to deal with the two categories of MHC-I loss defined above, and the existing evidence to support each strategy.

### 4.1. Restoration of pMHC-I Expression on Tumor Cells

MHC-I expression is frequently low or negative on tumor cell lines but upregulated rapidly in response to IFNγ [74,207,208,209,210]. Similarly, systemic administration of IFNγ to patients with a subset of “cold” sarcomas induced tumor MHC-I expression and T-cell infiltration [211]. The level of tumor MHC-I expression in fresh biopsies reflects T-cell activity and consequently IFNγ production, as demonstrated by (i) loss of MHC-I expression in IFNγ-resistant cells [77], (ii) identification of IFNγ signaling pathway components as essential mediators of tumor sensitivity to immune-mediated killing, in global genome-scale screens [202,203,204] and (iii) an increase in longitudinal IFNγ gene signatures early during therapy with immune checkpoint inhibitors in responding melanoma patients [75,212]. Immunotherapy with anti-PD-1 increases MHC-I expression on IFNγ-sensitive tumor cells in the presence of tumor-reactive T cells. As the success of ICB depends on T-cell recognition of tumor antigens, which in turn is determined by a limited number of responsive T-cell clones [137,213], therapies such as anti-CTLA-4 that broaden T-cell recognition of tumor antigens, increase the number of responding clones and synergize with anti-PD-1 in upregulating tumor MHC-I expression [214,215].

Inactivation of IFNγ signaling in tumor cells represents a major challenge to this approach and confers resistance to ICB [63,65,76,79]. Activation of type I interferon signaling provides an alternative pathway to the upregulation of MHC-I expression in IFNγ-resistant tumor cells [63,79,216], via treatments such as cGAMP (2′3′ cyclic guanosine monophosphate-adenosine monophosphate) and BO-112 (a nanoplexed version of polyinosinic:polycytidylic acid) that activate innate interferon sensing in tumor cells in the NLRC5- and IFNγ- independent manner [63,78]. PD-1 blockade delivered together with intratumoral BO-112 achieved partial responses in some patients with primary resistance to anti-PD-1 alone [217]. However, innate interferon sensing is often epigenetically inactivated in immunogenic tumors, via hypermethylation of the promoters of cyclic GMP-AMP synthase (*CGAS*) and stimulator of interferon genes (*STING1*) genes [218]. Impaired tumor STING signaling results in impaired immunogenicity and poor response to immunotherapies [219,220]. The cGAS-STING pathway sensitivity and MHC-I expression in tumor cells, as well as the resulting T-cell reactivity, could be restored by the DNMT inhibitors such as azacytidine [218]. Treatment with another DNMT inhibitor (guadecitabine) improved T-cell responses in the animal model of breast cancer via upregulation of tumor MHC-I expression and increased T-cell recruitment to the tumor [221]. In this study, an increase in MHC-I gene expression was also observed in a small cohort of breast cancer patients (*n* = 5) treated with a combination of the DNMTi azacytidine and the HDACi entinostat [221]. Inhibition of HDACs also increases tumor MHC-I expression. Drugs such as vorinostat, panobinostat and belinostat (targeting HDAC classes II, III and IV) and romidepsin (HDAC1/2 inhibitor) are currently approved for use in hematological malignancies (reviewed in [222]). In solid tumors, HDAC inhibition increased tumor MHC-I expression and T-cell recognition in Merkel cell carcinoma [223], glioma [111,224] and melanoma [225]. Activating mutations in the *EZH2* gene are frequently observed in diffuse large B cell lymphoma and are strongly associated with the lack of MHC-I/MHC-II expression and reduced T-cell infiltration; treatment with EZH2 inhibitors (EZH2i) restored MHC expression by de-repression of *NLRC5* and *CIITA* promoters in *EZH2* mutant cells [44]. EZH2 also maintains *MHC-I* epigenetic silencing in a subset of solid tumors including small cell lung cancer, neuroblastoma and Merkel cell carcinoma [43]; EZH2 inhibition restored MHC-I expression in lung cancer via *NLRC5* and *IRF1* de-repression [43]. EZH2 is often overexpressed in human melanoma [226], driving melanoma de-differentiation and loss of immunogenicity [113], and EZH2 inhibition synergized with immunotherapy in a mouse model of melanoma [113]. Finally, hypofractionated radiation synergized with immunotherapy by increasing MHC-I expression through both interferon-dependent and interferon-independent pathways [58,96,227].

In summary, monotherapy with anti-PD-1 increases tumor MHC-I expression via increased T-cell reactivity and IFNγ production, while the addition of anti-CTLA-4 broadens immune reactivity which may further increase MHC-I expression; activation of type I IFN signaling, epigenetic drugs (DNMTi, HDACi, EZH2i) and radiation synergize with immunotherapy to increase tumor MHC-I expression and immune recognition, also providing a pathway to improved T-cell recognition in IFNγ-resistant tumors.

### 4.2. Re-Directed Immune Recognition

Tumors that have lost MHC-I expression due to irreversible genetic alterations (mechanisms reviewed in Section 2), are no longer subject to CD8 T cell-mediated control. Therefore, immune recognition and immune-mediated control need to be re-directed to other immune subsets. The two immune cell subsets that can principally control MHC-I null tumor variants, are natural killer (NK) cells and CD4 T cells. The biology of NK cells and alterations of NK cell function in cancer are described in Section 4.2.1.

#### 4.2.1. Role of NK Cells in the Elimination of MHC-I Negative Cancers

The role of NK cells in cancer control was recognized over 40 years ago when increased growth and metastasis of transplantable tumors was demonstrated in mice with impaired NK cell activity [228] or following antibody-mediated NK cell depletion [229]. A decreased NK cell function was found in cancer patients and their families [230,231,232], and an increased risk for developing cancer was reported for healthy individuals with low activity of circulating NK cells [233]. NK cells constitute a high proportion of lymphocytes in the blood and at some tissue sites such as lung and liver [234,235]. NK cells have the innate ability to recognize and eliminate cells that have downregulated or lost MHC-I expression, via release of cytotoxic granules or through the engagement of the TNF-receptor superfamily death receptors on target cells with NK cell-expressed ligands such as tumor necrosis factor-related apoptosis-inducing ligand (TRAIL). Therefore, NK cells represent an attractive therapeutic target particularly for patients with MHC-I deficient cancers. NK cells rely on an array of germline-encoded receptors for target cell recognition (Figure 2A).

Circulating NK cells exhibit a diverse expression pattern of activating and inhibitory receptors, chemokine receptors, cytokines and cytolytic molecules [236], allowing NK cells to collectively display tolerance towards cellular targets expressing normal levels of MHC-I molecules while eliminating stressed cells that have down-regulated MHC-I and up-regulated activating ligands, such as MHC-I-related chain A (MICA) and MICB proteins in cancer [237,238] (Figure 2A).

The default inhibitory signal results from the engagement of NK cell inhibitory receptors by classical MHC-I molecules on target cells. The four killer cell immunoglobulin-like receptors (KIRs) recognize distinct groups of polymorphic MHC-I alleles, principally HLA-C (for example, KIR2DL1/CD158a reacts with the “C2 epitope group” of the HLA-C; Figure 2A). In addition to KIRs (or their Ly49 counterpart in mice), the inhibitory receptor CD94:NKG2A monitors the overall MHC-I expression via recognition of the non-classical HLA-E molecule loaded with a limited set of highly conserved peptides derived from leader sequences of classical MHC-I molecules, so that HLA-E expression reflects the total amount of MHC-I produced in a given cell [239]. Continuous interactions with self-MHC-I molecules during NK cell development, termed NK cell “education” or “licensing”, result in improved NK functionality [240,241]. Mature NK cells undergo functional adaptation in changed MHC-I environments; importantly this adaptation does not require cell division nor NK cell receptor acquisition or loss [242] and is achieved exclusively by means of tuning the cell surface expression of the inhibitory receptor(s) specific for the cognate MHC-I ligand [243]. In the context of cancer, this implies that (i) NK cells can downmodulate their expression of inhibitory receptors in the microenvironment of MHC-I deficient tumors, resulting in NK functionality loss, and that (ii) such newly “unlicensed” NK cells require a higher activation signal input to achieve a functional response. In the early 21st century, Velardi and colleagues provided the proof-of-principle that NK cells can eliminate cancer cells expressing a KIR inhibitory receptor-mismatched MHC-I allele while avoiding autoimmune side effects, in a setting of haploidentical hematopoietic transplantation for acute leukemias [244]. Following this pioneering work, immunotherapy with ex vivo expanded and activated allogeneic NK cells have been successfully used in the treatment of hematological malignancies, principally acute myeloid leukemia [245,246,247,248]. However, despite several ongoing clinical trials, it is not yet clear whether this model can also be expanded into solid cancers.

Several studies have suggested that NK cell infiltration into solid tumors was associated with favorable outcomes, such as in melanoma, renal and hepatocellular carcinoma [249,250,251,252]. The presence of NK cells in MHC-I-low melanoma lesions was associated with improved outcomes in the TCGA cohort [253], and NK cells were found in the proximity of melanoma cells in responders to anti-PD-1 treatment [254]. However, the results of these and other studies need to be interpreted with caution. NK cell inhibitory and activating receptors are not unique to NK cells as most are also expressed by CD8 T-cell subsets, with the exception of NKp46; furthermore, expression of CD56, CD16 and NKp46 (all routinely used as NK cell markers) is profoundly downregulated in tumor lesions [208,255,256,257,258]. This makes accurate NK cell identification (either by means of gene expression or immunohistochemistry) difficult. Multiple studies have demonstrated NK cell functional impairment in cancer. NK cells at the tumor site and in systemic circulation exhibited low levels of CD56 [259], activation receptors NKG2D, NKp40, NKp46, DNAM-1 [255,257,258,260,261,262,263], adaptor protein DAP12 [264] and CD16 [256], and displayed a profound reduction in cytolytic activity [258,263]. On the other hand, NK cells also displayed high levels of inhibitory checkpoints PD-1, PD-L1, LAG3, Tim-3 and TIGIT [251,256,265,266,267] and were capable of generating immunosuppressive adenosine and other soluble mediators [268,269] (Figure 2B). Additional mechanisms that help cancers avoid NK cell recognition and killing, are selective downregulation of MHC-I alleles involved in T-cell recognition (such as HLA-A2) while sparing KIR inhibitory ligands such as HLA-C, and proteolytic shedding of NKG2D ligands MICA and MICB [270,271,272,273] whose cell surface expression normally leads to tumor killing via NKG2D engagement [202,203] (Figure 2B). Indeed, high serum concentrations of MICA were associated with progression of neuroblastoma, melanoma, gastrointestinal, lung, breast, prostate, renal, gynecological cancers and multiple myeloma [274,275,276,277,278].

To test whether tumor MHC-I downregulation was associated with NK cell infiltration into tumors, we assessed intratumoral NK cells in a previously published cohort of 36 melanoma patients where both tumor and immune cells were comprehensively characterized by flow cytometry, including tumor cell surface MHC-I expression [34]. We found that CD244 was robustly expressed on tumor NK cells and used this marker to identify NK cells in viable melanoma tumors by flow cytometry (Figure 3a). Intratumoral NK cells were typically CD16-low, consistent with other studies (Figure 3a,b) and there was no relationship between tumor MHC-I expression and NK cell content, with effector CD8 T cells remaining predominant in MHC-I low tumors (Figure 3c).

Multiple therapeutic strategies have been proposed to stimulate NK cell responses in cancer (Figure 2C). Among them, blockade of proteolytic shedding of MICA or MICB through genetic engineering or via antibody-dependent metalloproteinase inhibition, restored potent NK cell-mediated killing tumor targets [270,271,279,280,281,282]. Increased tumor killing via antibody-mediated cellular cytotoxicity (ADCC), a potent mechanism reliant on the engagement of the Fc receptor CD16 by tumor-reactive antibodies (such as anti-CD20 antibody Rituximab used in B cell malignancies [283]), has also been reported for therapeutic antibodies targeting solid tumors, such as anti-HER2 Trastuzumab and anti-EGFR Cetuximab (clinical response however, was dependent upon additional adoptive transfer of high numbers of ex vivo expanded NK cells) [284,285,286]. Cell therapies with allogeneic NK cells or chimeric antigen receptor-engineered NK cells are ongoing (31 clinical trials with CAR NK cells were registered on clinicaltrials.gov in April 2021). Immunotherapy with anti-PD-1, anti-PD-L1, anti-Tim 3 and anti-TIGIT blocking antibodies was reported to improve NK cell function [287,288,289,290,291,292]. Finally, blockade of inhibitory receptors with anti-NKG2A antibody [monalizumab] synergized with PD-1 blockade in animal models and demonstrated an improved response rate in head and neck squamous cell carcinoma patients treated with a combination of anti-NKG2A and anti-EGFR, with the effect thought to have largely been mediated via CD8 T cells [293].

However, despite the multitude of therapeutic strategies aimed at NK cell activation, the precise interplay between activated NK cells and tumor-specific T cells remains unclear. On one hand, NK cell activation may inadvertently compromise T-cell activation and clonal expansion. A number of studies in animal models demonstrated a negative effect of NK cells on CD8 T cell-mediated anti-tumor responses. In transplantable animal models, NK cells suppressed CD8 T-cell priming and clonal expansion [294] through PD-1-dependent inhibition of antigen-presenting dendritic cells [295] and direct killing of recently activated T cells [296], while also attracting immunosuppressive regulatory T cells to the tumor [297]. On the other hand, T cell-derived IFNγ may prevent tumor killing by NK cells via enhanced MHC-I expression [253,298]. Inactivation of IFNγ signaling therefore sensitized tumors to NK-mediated killing [253,298]. Yet, while NK cells may play a non-redundant role in controlling MHC-I-null and IFNγ-resistant tumor variants, a higher incidence of such tumors in patients with acquired immunotherapy resistance signifies an escape from NK cell-mediated control. However, given a dramatic prevalence of NK cells in blood compared to the tumor tissue, it is tempting to speculate that NK cells may primarily be responsible for the control of MHC-I deficient and IFNγ-resistant clones that make their way into the systemic circulation. Indeed, the number of lung metastases formed by MHC-I deficient (and therefore, NK cell-sensitive) tumor cells was significantly lower compared with MHC-I sufficient clones, in a mouse model of metastatic melanoma [253]. Garrido et al. provide some clinical evidence suggesting that NK cells were capable of controling hematogenic, but not lymphatic spread of MHC-I deficient tumor clones in melanoma and colon cancer patients [206], while leukemia cells were found to frequently downregulate HLA-A and HLA-Bw6 alleles while selectively retaining the HLA-Bw4 alleles that interact with NK inhibitory receptors [299]. It is yet unclear whether NK cell-mediated metastasis-control mechanisms operate in advanced metastatic diseases in cancers with significant interlesional heterogeneity such as melanoma [300].

Of note, subsets of effector/memory TCRαβ CD8 T cells and TCRγδ T cells carry NK cell activating and inhibitory receptors and may be subject to activation via therapeutic manipulations described in this section.

#### 4.2.2. Role of CD4 T Cells in the Elimination of MHC-I Negative Cancers

The role of CD4 T cells as the “helpers” required for the appropriate stimulation of tumor-reactive CD8 T cells, has always been acknowledged (reviewed in [301]). CD4 T cells promote both effector and memory functions in CD8 T cells, including the provision of the appropriate co-stimulation and IL-2 required for T-cell priming and proliferation [302]. Immunogenic tumors often contain a mixture of reactive CD8 and CD4 T-cell clones [303], and neoantigen reactive CD4 T cells were common in metastatic melanoma [215]. In addition, melanoma, colon and breast cancer lesions were all enriched for mutations predicted to bind MHC-II molecules [304], and this was recently demonstrated for 6 cancer cell lines by the analysis of MHC-II ligandome using a proteomics approach [305]. CD4 T cells in expanded TIL products mediated objective responses in selected patients who received adoptive T-cell therapy [306,307,308]. In case studies, immunotherapy responses could be attributed to CD4 T cells, such as in a patient with metastatic cholangiocarcinoma who responded to adoptive transfer of neoantigen-specific CD4 TILs that directly recognized and killed tumor cells [177], or in a patient with a *BRAF*-mutant acral melanoma who sustained a complete clinical response after transfer of BRAF^V600E^-reactive CD4 TILs [309]. Although such reports are uncommon, they highlight the fact that CD4 T cells with a cytolytic potential infiltrate tumors, can be expanded for cellular therapy and mediate objective responses in cancer patients after adoptive transfer [177,309]. CD4 TILs reactive against hot-spot driver mutations were highly enriched in some patients with metastatic epithelial cancer and serous ovarian cancer [310]. In a vaccination setting, a higher CD4 T-cell response after vaccination with a synthetic peptide vaccine directed against human papillomavirus, was associated with a complete response and viral clearance in women with high-grade vulvar intraepithelial neoplasia [311]. In a small clinical trial with six melanoma patients, immunization with a personalized neoantigen-based synthetic vaccine induced polyfunctional CD4 T-cell responses more frequently than CD8 T-cell responses (60% versus 16%, respectively), which were associated with a complete response in four patients, while the two patients who progressed subsequently responded to immune checkpoint blockade with anti-PD-1 [185]. Finally, two recent reports showed that a cytotoxic CD4 T-cell gene signature was associated with immunotherapy response in bladder cancer [312] and that mismatch-repair deficient, MHC-I low tumors were highly infiltrated with CD4 T cells [313].

Clinical studies demonstrate that CD4 T cells with anti-tumor reactivity can mediate their effects via direct cytotoxicity against tumor targets [177,309,312], and by broadening the CD8 T-cell response in the context of immunotherapy, such as with anti-CTLA-4 antibodies [214]. Animal studies indicate that CD4 T cells can operate via these and additional mechanisms.

CD4 T cells help CD8 T cells kill tumor cells. Forced expression of MHC-I and MHC-II restricted antigens in the same tumor cell achieved immunotherapy response in an MHC-II negative, immunotherapy resistant model [314]. CD4 T-cell help was critical to achieve the optimal expansion and cytolytic activity of tumor-reactive CD8 T cells in the context of vaccination with tumor neoantigens, or immune checkpoint blockade with anti-PD-1 and/or anti-CTLA-4 [314].CD4 T cells can kill tumor cells directly. Although this function is usually attributed to CD8 T cells, tumor-reactive CD4 T cells can kill tumor cells expressing pMHC-II complexes, under certain conditions. Adoptively transferred, naïve transgenic CD4 T cells with anti-melanoma reactivity acquired the ability to control autologous tumors after differentiating into cytolytic effectors [315,316,317]. In these experiments, CD4 T-cell priming boosted by lymphopenia (achieved via irradiation, cytotoxic drug treatment or immunodeficient host environment) supported CD4 T-cell differentiation into IFNγ-producing, cytolytic effector cells that infiltrated tumors and killed their targets directly [315,316]. This mechanism, reliant on granzymes, perforin and IFNγ, has recently been reported to also operate in human tumors [312].CD4 T cells can control tumor growth by acting on tumor microenvironment. In experiments where direct interactions between CD4 T cells and their targets were excluded by MHC-II constraints (MHC-II negative tumor cells, or MHC-II allele ignorant CD4 T cells), adoptively transferred naïve transgenic CD4 T cells specific for tumor antigens, indirectly controlled growth of subcutaneous tumors when a correct MHC was provided on host-derived stromal cells [318,319,320,321]. This indirect killing largely relied on the production of IFNγ by T cells, as confirmed by treatment with IFNγ blocking antibodies or using animals deficient in IFNγ signaling [315,316,318]. The release of IFNγ into the tumor microenvironment has multiple effects, such as the destruction of tumor vasculature [322] and augmentation of macrophage-mediated killing [323,324]. Re-polarization of tumor macrophages towards the M1 phenotype resulted in the secretion of nitric oxide that penetrated tumor cells and triggered apoptosis through the formation of cytotoxic peroxynitrate [323,324]. CD4 TILs also produced tumor necrosis factor (TNF)α [318] that induced tumor cell senescence [325]. In agreement with the importance of indirect recognition of tumor-derived antigens for tumor control, a recent proteomics study on MHC-II ligandome in human cancer has suggested that intratumoral pMHC-II presentation is dominated by professional antigen-presenting cells rather than cancer cells [305].CD4 T cells can control tumor growth by cooperating with CD8 T cells to kill tumor stroma. In experiments where all direct interactions between T cells and their tumor targets were excluded by global MHC constraints (pan-MHC allele-ignorant T cells), tumor antigen-specific CD4 and CD8 T cells were both required to control tumors by targeting stromal cells that expressed the required MHC alleles [326].CD4 T cells provide B-cell help for the production of class-switched tumor-reactive antibodies, mainly of the IgG2a (mice) or IgG1 isotype (humans) [327,328]. Such antibodies can bind to tumors and trigger indirect tumor killing via ADCC by engaging cytotoxic macrophages and/or NK cells [329]. While the role of B cells as inferred from the animal models remains somewhat controversial (that is, tumor immunity is enhanced in B cell-deficient animals but also attenuated upon B-cell depletion in the context of immunotherapy [330]), human studies support a positive role for B cells in cancer. B cell tumor infiltration was positively associated with prognosis in melanoma, head and neck and ovarian cancers [331,332,333]. Expression of B cell-activating factor (BAFF) in human cancers tracked with IFNγ signatures and was positively associated with the overall survival in the TCGA melanoma cohort [334]. Finally, the presence of B cells in the tumor was positively associated with immunotherapy response in melanoma, renal cell carcinoma and sarcoma [335,336,337].

However, there is a major hurdle—CD4 T-cell response in tumors may also lead to the differentiation of suppressive regulatory T cells (Tregs), a subset of CD4 T cells that are defined by their expression of the transcription factor Foxp3. Systemic accumulation of Tregs in tumor-bearing mice and cancer patients is associated with poor prognosis [338], Treg depletion in animal models allowed tumor control [339], while the blockade of Treg function with anti-CTLA-4 antibodies allowed tumor control in animal models [329,340,341] and in a subset of patients with advanced melanoma [342]. The precise molecular pathways that guide differentiation of tumor antigen-specific CD4 T cells into cytolytic effectors versus regulatory T cells are still poorly understood, but animal studies indicate that neoantigen expression leads to tumor regulatory T-cell accumulation [343], and a large proportion of immunologically active tumors contain actively proliferating Tregs alongside CD8 T cells and conventional, i.e., Foxp3-negative CD4 T cells. While immunotherapy with anti-PD-1 may paradoxically enhance Treg-dependent suppression [344], targeting CTLA-4 either depletes this subset or blocks regulatory T-cell activity [329,345]. Thus, anti-CTLA-4 has to be considered as part of therapeutic strategies directed at CD4 T cells.

## 5. Concluding Remarks

Quantitative protein expression data indicate that tumor cell surface expression of MHC-I molecules is an important indicator of immunotherapy sensitivity [15,34]. Where such quantitative analysis is possible (e.g., flow cytometry in freshly dissociated tumors [34]), we propose that anti-PD-1 monotherapy is selected as first-line immunotherapy for tumors with normal/high MHC-I expression, while alternative approaches are used for MHC-I^low^ tumors (Figure 4). MHC-I^low^ tumors are further subdivided into either “soft” or “hard” lesions, depending on the integrity of the IFNγ pathway (intact/deficient), which could be established by a brief in vitro exposure to IFNγ [210]. For tumors with IFNγ intact signaling, use of anti-PD-1 plus epigenetic modifiers is warranted to allow de-repression of MHC-I genes, with a potential additional benefit of [enhanced immune recognition as the result of] increased neoantigen expression [346]. For IFNγ resistant tumors, the use of BO-112, fractionated radiation or oncolytic viruses will allow restoration of MHC-I expression [by triggering innate interferon signaling] [58,78,217,219,220,227] and potentially response to PD-1 blockade; epigenetic drugs may also be used to boost the response. Finally, for IFNγ non-responsive, MHC-I^null^ (“hard”) lesions, combination immunotherapy with anti-PD-1 plus anti-CTLA-4 is warranted. The rationale here is to re-direct the response towards CD4 T cells (with the use of anti-CTLA-4 helping attenuate Treg-mediated suppression), plus NK-targeting drugs (ICB or antibodies preventing NKG2DL shedding) to prevent hematogenic metastatic spread [281,282]. In this setting, therapeutic blockade of PD-L1 will help unleash the reactivity of PD-L1-expressing NK cells against MHC-I^low^ tumors [290], while also improving antigen presentation in the tumor microenvironment [347]. This approach described in this section is suitable for tumors amenable to quantitative flow cytometry as described in [34].

## Figures and Tables

**Figure 1 ijms-22-06741-f001:**
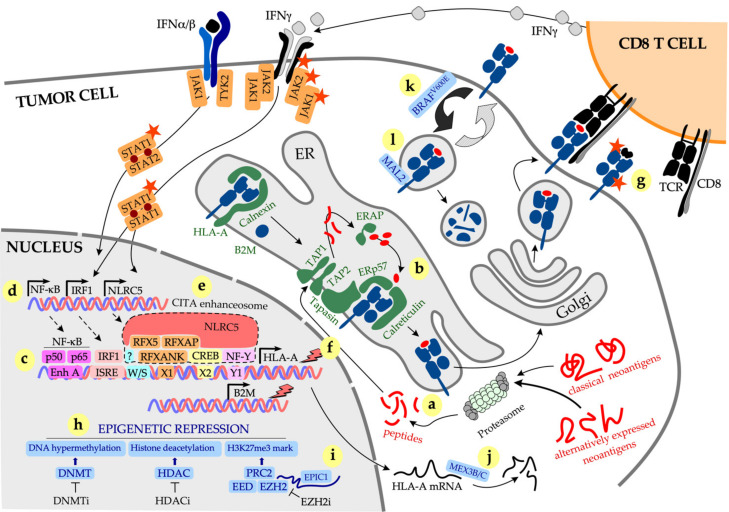
Regulation and alterations of MHC class I expression in cancer. Peptide-MHC class I (pMHC-I) complexes are assembled in the endoplasmic reticulum (ER) and transported to the cell surface via Golgi and secretory vesicles. (**a**,**b**) Classical or alternate neoantigen peptides are generated in the cytosol by the proteasome (**a**), transported to the ER by transporter associated with antigen processing (TAP1 and 2), trimmed by ER aminopeptidases (ERAP) to the optimal size and loaded onto MHC-I molecules with the help of antigen processing machinery complex consisting of tapasin, ER protein 57, and calreticulin (**b**). (**c**) Expression of MHC-I heavy chains is subject to regulation by transcription factors binding to multiple sites in the *MHC-I* promoter region. (d) Expression of the nuclear factor kappa-light-chain-enhancer of activated B cells (NF-kB), interferon regulatory factor 1 (IRF1) and the master regulator of MHC-I expression, Nucleotide-binding oligomerization domain-Like Receptor family Caspase recruitment domain containing 5 (NLRC5) is upregulated by interferons, in particular IFNγ, produced by T cells after pMHC-I recognition. (**e**) NLRC5 assembles the class I transactivator (CITA) enhanceosome that binds to four distinct sites (W/S, X1, X2, Y1 boxes) within the *MHC-I* promoter. (**f**) *MHC-I* expression is affected by mutations and deletions in the *MHC-I* heavy chains and beta-2 microglobulin (*B2M*) genes (indicated by lightning), as well as loss-of-function mutations in the components of the IFN signaling complex, such as in IFNγ receptor, the associating adaptors Janus kinases 1 and 2 (*JAK1/2*) and the transcription factor Signal transducer and activator of transcription 1 (*STAT1*) (mutations in proteins are indicated with stars). (**g**) pMHC-I recognition by CD8 T cells is also affected by mutations in the α3 domain [CD8 co-receptor binding] and α1/ α2 domains [peptide-binding groove; spectrum of displayed peptides]. (**h**) MHC-I expression is subject to epigenetic regulation, with DNA hypermethylation, histone deacetylation and trimethylation of histone 3 lysine 27 (H3K27me3) by polycomb repressive complex 2 (PRC2) all resulting in a dramatic reduction of the MHC-I expression. Accordingly, inhibition of DNA Methyltransferases (DNMTi), Histone Deacetylases (HDACi) and the catalytic subunit of the PRC2 complex, Enhancer of Zeste Homolog 2 (EZH2i), increase pMHC-I expression on the surface of tumor cells. (**i**–**k**) Other mechanisms that negatively regulate MHC-I cell surface expression: epigenetically activated long non-coding (lnc)RNA *EPIC1* interacts with and activates EZH2 (**i**), mRNA-binding proteins MEX3B and MEX3C destabilize *HLA-A2* mRNA (**j**), mutant BRAF^V600E^ promotes internalization and intracellular sequestration of MHC-I (**k**), MHC-I binding protein Myelin and lymphocyte protein 2 (MAL2) promotes MHC-I endocytosis and degradation (**l**). Abbreviations: BRAF, v-raf murine sarcoma viral oncogene homologue B1; DNMTi, DNA methyltransferase inhibitor; CREB, cAMP response element-binding protein; Enh, enhancer; EZH2, enhancer or zeste homolog 2; H3K27me3, trimethylation of histone 3 lysine 27; HDACi, histone deacetylase inhibitor; NF-Y, nuclear transcription factor Y; MEX3B, Mex3 RNA-binding family member B; RFX5, regulatory factor X5; RFXANK, regulatory factor X associated ankyrin containing protein; TSA, tumor-specific antigen.

**Figure 2 ijms-22-06741-f002:**
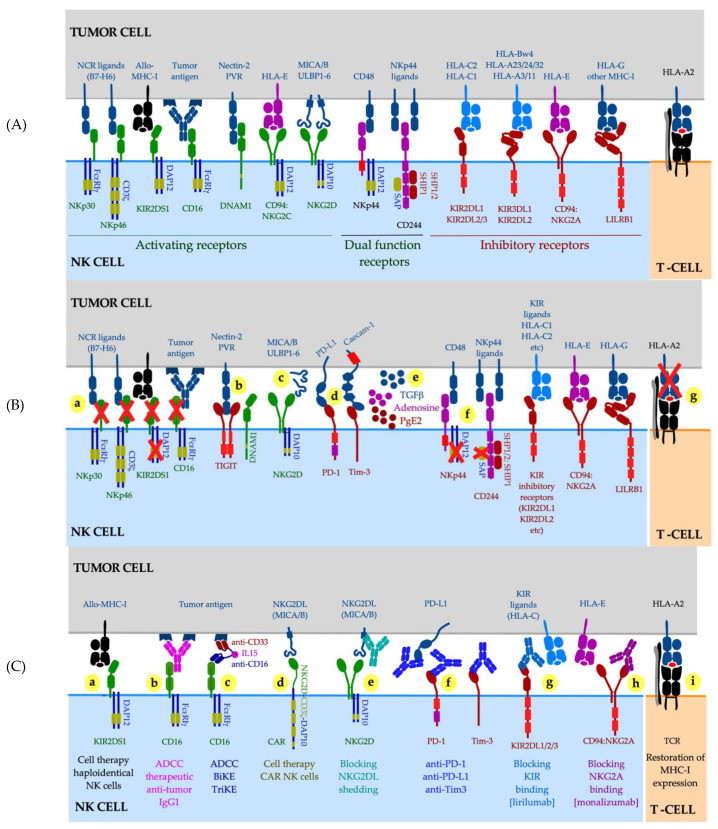
Alteration of NK cell functions in cancer and strategies to overcome it. (**A**) NK receptors and their principal ligands relevant to oncoimmunology. Activating receptors are shown in green, the associating signaling adaptors are shown in blue, with immunoreceptor tyrosine-based activation motifs (ITAMs) marked in olive. Inhibitory receptors with immunoreceptor tyrosine-based inhibition motifs (ITIM) are shown in red. Dual function receptors (purple) provide either activating or inhibitory input, via a combination of ITAMs or ITIMs (NKp44) or through immunoreceptor tyrosine-based switch motifs (ITSMs) that recruit both activating adaptors and phosphatases (CD244). (**B**) Mechanisms preventing efficient NK cell activation in cancer: downregulation of activating receptors and/or adaptors (a); engagement of DNAX accessory molecule (DNAM)1 ligands by the inhibitory competitor T-cell immunoglobulin and ITIM domain (TIGIT) (b); proteolytic shedding of NK group 2D (NKGD) ligands MHC-I chain-related proteins (MIC)A and MICB (c); expression of inhibitory checkpoints (d); production of inhibitory mediators adenosine, prostaglandin E2 and transforming growth factor beta (TGFβ) (e); switch to inhibitory signaling for dual function receptors (f); selective loss of MHC-I alleles relevant for T-cell recognition with retention of MHC-I alleles engaging KIR inhibitory receptors (g). (**C**) Therapeutic strategies directed at stimulating NK cell responses in cancer: haploidentical hematopoietic stem cell transplantation or NK cell therapy (a); triggering antibody-dependent cellular cytotoxicity (ADCC) with tumor-reactive therapeutic antibodies of the IgG1 class that bind CD16 (b); triggering ADCC with bispecific and tri-specific killer engagers (BiKE, TriKE) that bind to both tumor antigen and CD16 (c); introduction of chimeric antigenic receptors (CAR)s into NK cells through genetic engineering, followed by ex vivo expansion and adoptive transfer (d); antibody-mediated blockade of proteolytic shedding of NKG2D ligands (NKG2DL) (e); immune checkpoint blockade with anti-Programmed Death (PD)-1, anti-PD-L1 or anti-Tim3 antibodies (f); disruption of MHC-I-mediated NK cell inhibition with the KIR-blocking antibody lirilumab (g) or NKG2A-blocking antibody monalizumab (h); redirection of response towards CD8 T-cells by boosting MHC-I expression on tumors, such as with epigenetic drugs and genotoxic therapies (i). Note that (f-h) affect both NK cells and CD8 T-cells. Abbreviations used: Caecam-1, carcinoembryonic antigen-related adhesion molecule-1; DAP12, DNAX activation protein of 12kDa; KIR2DL, killer cell immunoglobulin-like receptor, two immunoglobulin domains and long cytoplasmic tail; KIR2DS1, KIR with two immunoglobulin domains and short cytoplasmic tail 1; LILRB1, leukocyte immunoglobulin-like receptor subfamily B member 1; NKG2A, NK group 2A; PgE2, prostaglandin E2; PVR, Poliovirus receptor; SAP, signaling lymphocytic activation molecule-associated protein; SHP, the Src homology region 2 (SH2) domain-containing phosphatase; SHIP, SH2 domain-containing inositol phosphatase; TCR, T-cell receptor; TGFβ, Transforming growth factor beta; ULBP, UL-16 binding protein.

**Figure 3 ijms-22-06741-f003:**
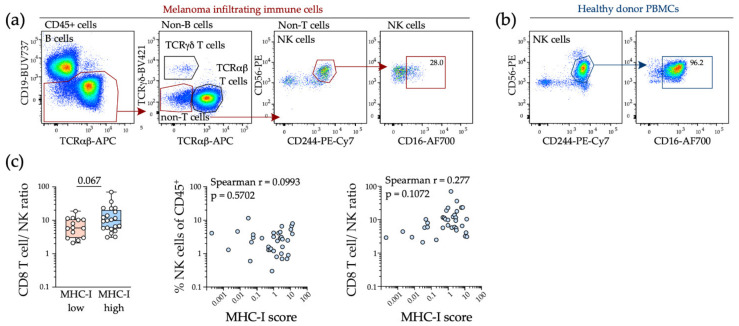
Abundance of tumor-infiltrating NK cells in immunotherapy-naïve melanoma patients does not correlate with tumor MHC-I expression or loss. (**a**) Identification of tumor-infiltrating NK cells by flow cytometry. Left to right, NK cells were identified as CD45^+^ (not shown) CD19^−^, TCRαβ ^−^TCRγδ^−^, CD244^+^CD56^+^ cells. Expression of CD56, CD244 and CD16 is shown for tumor and control blood (**b**). (**c**) Lack of correlation between melanoma MHC-I expression and NK cell infiltration. Left to right, ratio of (CD45RO^+^TCRαβ^+^CD8^+^) TILs to NK cells in MHC-I^high^ (MHC-I score ≥ 1.0) and MHC-I^low^ melanoma biopsies (MHC-I score <1.0); correlation of NK cell frequency and melanoma MHC-I expression score; correlation of CD8/NK cell ratio and melanoma MHC-I expression score. For the details on the study cohort (*n* = 36) and the calculation of the tumor MHC-I score, see reference [34].

**Figure 4 ijms-22-06741-f004:**
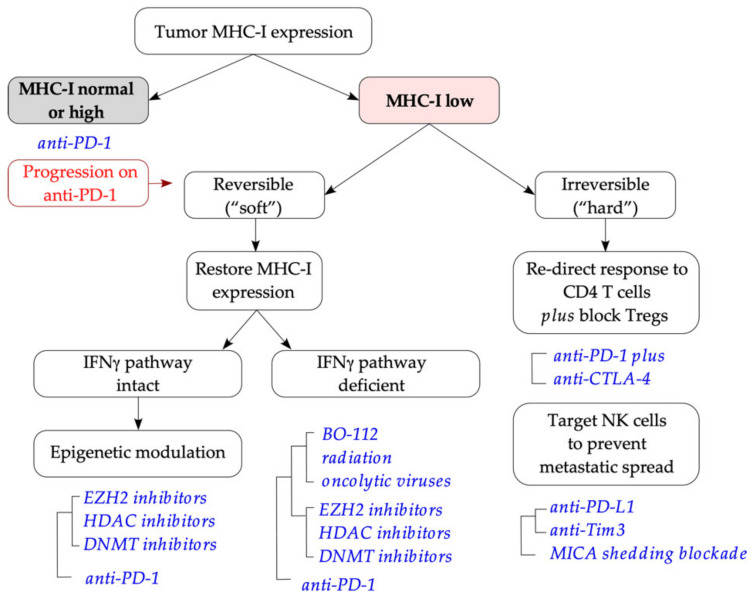
Proposed strategies of immunotherapy selection based on tumor MHC-I cell surface expression. Patients with adequate tumor cell surface MHC-I expression are selected for the first line anti-PD-1 monotherapy (progression while on therapy warrants consideration of other strategies). For MHC-I low tumors, the strategy depends on whether the deficit is reversible (e.g., transcriptional downregulation) or not (e.g., gene deletion). For reversible alterations, MHC-I expression is restored; for tumors with intact IFNγ signaling and altered epigenetic status, a combined strategy (epigenetic inhibitors plus anti-PD-1) is warranted; for tumors deficient in IFNγ signaling, MHC-I expression is restored via bypassing the IFNγ pathway with innate interferon sensing (e.g., BO-112, fractionated radiation or oncolytic viral therapy, plus anti-PD-1). For irreversible MHC-I loss, the immune response is re-directed to CD4 T cells and NK cells. Combined immunotherapy is used to activate conventional CD4 T cells while also blocking regulatory T cells with anti-CTLA-4; activation of NK cells (immunotherapy and engagement of activating receptors) is used to prevent hematogenic metastatic spread.

**Table 1 ijms-22-06741-t001:** Types of tumor antigens and methods of analysis and/or prediction.

Antigen Type (Example)	Prediction Pipeline	Analysis	Method and/or Tool	Central Tolerance	Reference
TAA, overexpressed (HER2)	Identification Validation	Gene expression, protein expression Immunological assays	RNAseq NGS IHC, FC T-cell co-culture assays	Yes	[120,121,122]
TAA, tissue differentiation (tyrosinase)	Identification Validation	Gene expression, protein expression Immunological assays	RNAseq NGS IHC, FC T-cell co-culture assays	Yes	[121,122,123]
TAA, cancer germline (NY-ESO-1)	Identification Validation	Gene expression, protein expression Immunological assays	RNAseq NGS IHC, FC T-cell co-culture assays	No	[121,122]
TIEPP (CALCA)	Identification Validation	Protein expression Immunological assays	Computational T-cell co-culture with TAP-deficient targets	No	[124,125,126]
TSA, hERV(HERV-K)	Identification Validation	Gene expression Immunological assays	RNAseq NGS T-cell co-culture assays	No	[121,122,127]
TSA, oncoviral (HPV16 E6/E7)	Molecular diagnostics Validation	Genotyping Immunological assays	DNA hybridization, PCR T-cell co-culture assays	No	[127,128]
TSA, classical neoantigens (unique sequence)	**Computational** **prediction**		**Tools**	No	
	1. HLA typing	DNA WGS, WES; RNAseq-NGS	OptiType ^1^, Polysolver ^1^		[52,120,129]
	2. Inference of mutated peptides	DNA WGS, WES; RNAseq-NGS	pVAC-seq		[130,131]
	3. In silico prediction of HLA binding	Computational	NetMHCpan NetMHCpan-4.0 NetMHCIIpan		[132,133]
	4. In silico prediction of antigen presentation	Computational	Netchop, NetCTL		[134,135]
	5. Candidate neoantigen prioritization	Computational; Immunopetidomics	Combined prediction tools; MS/MS		[122,136,137]
	6. Validation	Computational; Immunological assays	pMHC-TCR models (MODELLER, LYRA) T-cell reactivity with peptide-HLA multimers; T-cell co-culture with autologous or HLA-matched targets expressing selected neoantigens or a library		[138,139]
aeTSA, cryptic neoantigens (unique sequence)	**Proteogenomics**			No	[140,141,142,143,144]
	1. HLA ligandome	pMHC isolation, peptide identification	HLA immunoaffinity purification; MS/MS		
	2. Global transcription	RNAseq	RNAseq NGS		
	3. Ribosome profiling	Ribo-seq	Ribo-seq NGS		
	4. Dataset narrowing	Translation efficiency and library filtration	Translation efficiency (Ribo-seq/RNAseq counts); filtration (tissue and mTEC libraries)		
	5. Validation	Immunological assays	T-cell co-culture with autologous or HLA-matched targets expressing selected neoantigens or a library		

^1^ Only identify binding to HLA-A and HLA-B alleles. Abbreviations: aeTSA, alternatively expressed tumor-specific antigens; CALCA, calcitonin-related polypeptide alpha; FC, flow cytometry; HER2, human epidermal growth factor receptor 2; hERV, human endogenous retroviruses; HPV, human papillomavirus; IHC, immunohistochemistry; mTEC, medullary thymic epithelial cells; MS/MS, tandem mass spectrometry; PCR, polymerase chain reaction; TAA, tumor-associated antigens; TIEPP, T-cell epitopes associated with impaired peptide processing; TSA, tumor-specific antigens; NGS, next-generation sequencing; WES, whole-exome sequencing; WGS, whole-genome sequencing.

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
