# Peer review of "MHC Class I Deficiency in Solid Tumors and Therapeutic Strategies to Overcome It"

_ijms, 2021, doi:10.3390/ijms22136741_

Round 1

Reviewer 1 Report

High quality review on the role of the MHC class I expression in tumors and the consequences for therapeutic approach. Well referenced.

I have only few (minor) remarks:

line 158, first sentence: use lettertype bold

line 197 "or" should be "of"

table I:

  • include whether central tolerance exists yes or no for each item. This seems important to me
  • "TSA(other viruses)" should be included for completeness, such as HPV induced tumors
  • TIEPP is not included in table. Please add. Could TIEPP not be classified as TSA rather than TAA?

Line 616-617. Please note that also subsets of TCRgammadelta and CD8 effector memory T cells carry NK receptors and that these cells can be activated via other mechanisms than via the TCR.

figure 4:it is unclear to me why PDL1 would be more effective than PD1 to prevent metastatic spread. Please present evidence.

Author Response

High quality review on the role of the MHC class I expression in tumors and the consequences for therapeutic approach. Well referenced.

We thank the reviewer for their careful consideration of our manuscript, favourable assessment of the quality of our work and their helpful comments. Please find the point-by-point reply to this reviewer’s comments below.

I have only few (minor) remarks:

line 158, first sentence: use lettertype bold

line 197 "or" should be "of"

The formatting and the typographical error have been corrected.

table I:

  • include whether central tolerance exists yes or no for each item. This seems important to me
  • "TSA(other viruses)" should be included for completeness, such as HPV induced tumors
  • TIEPP is not included in table. Please add. Could TIEPP not be classified as TSA rather than TAA?

We have substantially modified Table 1 and the relevant sections of the text so that the two are now better aligned. Table 1: “Central tolerance”, “TIEPP” and “oncoviral” sections have been added, as well as examples for each antigen category. “Oncogenic viruses” are now described in section 3.2 (lines 414-421). We have also added some information about TIEPP antigens, highlighting their relevance to immune responses in general and human disease in particular (lines 385-389). With regards to the last comment, we prefer to classify TIEPP as TAA rather than TSA, as they are not mutated.

Line 616-617. Please note that also subsets of TCRgammadelta and CD8 effector memory T cells carry NK receptors and that these cells can be activated via other mechanisms than via the TCR.

We have now included this information at the conclusion of section 4.2.1 (lines 803-805)

figure 4:it is unclear to me why PDL1 would be more effective than PD1 to prevent metastatic spread. Please present evidence.

A recent study demonstrated that PD-L1 blockade could control PD-L1-negative tumors in vivo by activating PD-L1-expressing NK cells (Dong Cancer Disc 2019, 9:1422). We have also previously discussed the various roles of PD-L1 expressed in the tumour microenvironment (Shklovskaya and Rizos, IJMS 2020, 21:7139). We have added an explanation and relevant references to section 5 (lines 929-931).

Reviewer 2 Report

In the present study, authors present an extensive review on MYC-I deregulation in tumors, the relative immune escape mechanisms developed by cancer cells and potential therapeutic strategies to overpass it.

The review is well written and nicely constructed to separate sections that lead the readership to comprehend the significance of MYC-I in immunity and how deregulation/interference of the pMHC-I complex formation may be part of the tumor escape mechanisms, benefit tumor progression and lead to resistance to administered immunotherapies.  In particular, authors describe in detail the various mechanisms that may lead to MHC-I loss of expression, the distinct categories of antigens that may trigger immune responses and finally the different therapeutic approaches that have been implied to overcome the MHC-I downregulation separated on the basis of MHC-I expression reversibility via either MHC-I restoration or alternative ΝΚ or CD4 mediated immune cascades.

I have only a few minor comments:

  • Table 1 should agree with what is mentioned on the text. For example in Lines 330-332 authors mention that TAAs are separated in four categories, whereas in Table 1 there are only three. In this case, maybe the oncofetal TAAs (e.g. CEA) should be mentioned in a separate capture in the text and Table should include T cell epitopes associated with impaired peptide processing (TIEPP) in a separate category of TAAs.Abbreviations for hERV and aeTSA should be included in the abbreviation list of Table 1.

  • Authors should also include a category or comment on differentiation antigens referring to a particular phase of a cell type differentiation (e.g. NY-BR-1).

  • Line 397: Tumor mutation burden should be written upfront TMB which is now explained in line 426.

  • Lines 445-469: Authors describe existing difficulties and present relevant explanations for weaknesses in accurately predicting response to applied immunotherapy for each patient. The majority of cancer types are characterized by profound innate heterogeneity on the genetic/molecular level, the epigenetic features but also on the tumor microenvironmental features. Authors should comment on that and include this parameter in the ones already stated.

Author Response

In the present study, authors present an extensive review on MYC-I deregulation in tumors, the relative immune escape mechanisms developed by cancer cells and potential therapeutic strategies to overpass it.

The review is well written and nicely constructed to separate sections that lead the readership to comprehend the significance of MYC-I in immunity and how deregulation/interference of the pMHC-I complex formation may be part of the tumor escape mechanisms, benefit tumor progression and lead to resistance to administered immunotherapies.  In particular, authors describe in detail the various mechanisms that may lead to MHC-I loss of expression, the distinct categories of antigens that may trigger immune responses and finally the different therapeutic approaches that have been implied to overcome the MHC-I downregulation separated on the basis of MHC-I expression reversibility via either MHC-I restoration or alternative ΝΚ or CD4 mediated immune cascades.

We thank the reviewer for their careful consideration of our manuscript and helpful comments. Please find the point-by-point reply to this reviewer’s comments below.

I have only a few minor comments:

Table 1 should agree with what is mentioned on the text. For example in Lines 330-332 authors mention that TAAs are separated in four categories, whereas in Table 1 there are only three. In this case, maybe the oncofetal TAAs (e.g. CEA) should be mentioned in a separate capture in the text and Table should include T cell epitopes associated with impaired peptide processing (TIEPP) in a separate category of TAAs. Abbreviations for hERV and aeTSA should be included in the abbreviation list of Table 1.

We have modified Table 1 and the relevant section(s) of the text in line with this reviewer’s comments (Reviewer 1 also pointed out inconsistencies). “TIEPP antigens” and “oncoviral antigens” have been added to Table 1, and the corresponding sections of the text expanded (TIEPP antigens, lines 385-389; oncoviral antigens, lines 414-421). We have also included examples for each category of antigens and expanded the abbreviations list for Table 1, as requested by this reviewer.

Authors should also include a category or comment on differentiation antigens referring to a particular phase of a cell type differentiation (e.g. NY-BR-1).

NY-BR-1 has been added to “Tissue differentiation antigens” section (lines 355-359)

Line 397: Tumor mutation burden should be written upfront TMB which is now explained in line 426.

The abbreviation has been corrected (line 408)

Lines 445-469: Authors describe existing difficulties and present relevant explanations for weaknesses in accurately predicting response to applied immunotherapy for each patient. The majority of cancer types are characterized by profound innate heterogeneity on the genetic/molecular level, the epigenetic features but also on the tumor microenvironmental features. Authors should comment on that and include this parameter in the ones already stated.

A statement to this effect has been included (lines 489-490)

Reviewer 3 Report

In this review, Shklovskaya and Rizos have described a state of art of MHC structure and function with a focus on tumor escape mechanisms to downregulate its function. The manuscript also describes the therapeutic strategies directed at overcoming tumor MHC-I downregulation or loss. In particular, the authors describe the current strategies to restore MHC-I expression and function, but they also define MHC-I-independent immunotherapeutic strategies based on CD4 T cells and NK cells.

The manuscript is well written and gives a good overview of the issue.

The authors should provide minor revisions which will render the review exhaustive and suitable for publication in IJMS.

  1. The authors discuss the importance of MHC-I/antigen affinity. I suggest to point up the concept of stability that in some cases has been described as a better predictor for antigen immunogenicity. I suggest to cite the paper from M. Harndahl et al 2012 that deeply defines this evidence “Peptide-MHC class I stability is a better predictor than peptide affinity of CTL immunogenicity” or similar articles.
  2. Line 12 (abstract): “MHC-I loss can render tumor cells invisible to the immune system”. I suggest to specify that MHC-I loss render tumor invisible to T cells “MHC-I loss can render tumor cells invisible to cytotoxic T cells”.
  3. Section 2.1 line 53: If I understand well, the authors refer to trimer composed by B2m, MHC-I heavy chain and antigen. I suggest to replace “short peptide” with “short peptide antigen” to avoid misunderstandings.
  4. Section 2.2 line 158: “Germline MHC-I diversity” I think it should be bold.
  5. Section 4.1 line 583: In the field of STING and MHC, I suggest to include oncolytic viruses in the repertoire of therapeutics able to restore MHC expression via STING dependent mechanisms. In my opinion the two following papers should be cited: "STING signaling in melanoma cells shapes antigenicity and can promote antitumor T-cell activity" and "Integrity of the Antiviral STING-mediated DNA Sensing in Tumor Cells Is Required to Sustain the Immunotherapeutic Efficacy of Herpes Simplex Oncolytic Virus".
  6. Section 4.2.2 line 843: It could be of interest to discuss the recent paper from Bardelli describing the role of CD4 T cells in ICI response in B2m deficient tumors "CD4 T cell dependent rejection of beta 2 microglobulin null mismatch repair deficient tumors".
  7. Section 6 line 896: this concept should be expanded to all the agonists of PRRs to include RNA sensing (RNA oncolytic viruses, BO-112), DNA sensing (DNA oncolytic viruses, STING agonists) and TLRs.
  8. Figure 4: PRR agonists could be included among therapeutic strategies in “IFNg pathway deficient”

Author Response

In this review, Shklovskaya and Rizos have described a state of art of MHC structure and function with a focus on tumor escape mechanisms to downregulate its function. The manuscript also describes the therapeutic strategies directed at overcoming tumor MHC-I downregulation or loss. In particular, the authors describe the current strategies to restore MHC-I expression and function, but they also define MHC-I-independent immunotherapeutic strategies based on CD4 T cells and NK cells.

The manuscript is well written and gives a good overview of the issue.

The authors should provide minor revisions which will render the review exhaustive and suitable for publication in IJMS.

We thank the reviewer for their careful consideration of our manuscript and helpful comments. Please find the point-by-point reply to this reviewer’s comments below.

  1. The authors discuss the importance of MHC-I/antigen affinity. I suggest to point up the concept of stability that in some cases has been described as a better predictor for antigen immunogenicity. I suggest to cite the paper from M. Harndahl et al 2012 that deeply defines this evidence “Peptide-MHC class I stability is a better predictor than peptide affinity of CTL immunogenicity” or similar articles.

We have added a statement describing the importance of pMHC-I complex stability to section 3.3 (lines 531-533), as pMHC-I stability most significantly affects presentation of alternative TSAs.

  1. Line 12 (abstract): “MHC-I loss can render tumor cells invisible to the immune system”. I suggest to specify that MHC-I loss render tumor invisible to T cells “MHC-I loss can render tumor cells invisible to cytotoxic T cells”.

The entire sentence reads, “MHC-I molecules present antigenic peptides to cytotoxic T cells, and MHC-I loss can render tumor cells invisible to the immune system.” We believe that “cytotoxic T cells” should not appear twice in the same sentence in this particular context.

  1. Section 2.1 line 53: If I understand well, the authors refer to trimer composed by B2m, MHC-I heavy chain and antigen. I suggest to replace “short peptide” with “short peptide antigen” to avoid misunderstandings.

This has now been amended (line 53)

  1. Section 2.2 line 158: “Germline MHC-I diversity” I think it should be bold.

This has now been amended (line 159)

  1. Section 4.1 line 583: In the field of STING and MHC, I suggest to include oncolytic viruses in the repertoire of therapeutics able to restore MHC expression via STING dependent mechanisms. In my opinion the two following papers should be cited: "STING signaling in melanoma cells shapes antigenicity and can promote antitumor T-cell activity" and "Integrity of the Antiviral STING-mediated DNA Sensing in Tumor Cells Is Required to Sustain the Immunotherapeutic Efficacy of Herpes Simplex Oncolytic Virus".

We have amended the relevant section to include both references (lines 603-605)

  1. Section 4.2.2 line 843: It could be of interest to discuss the recent paper from Bardelli describing the role of CD4 T cells in ICI response in B2m deficient tumors "CD4 T cell dependent rejection of beta 2 microglobulin null mismatch repair deficient tumors".

We thank the reviewer for this suggestion. We have amended the relevant section to include this reference (line 836)

7-8. Section 6 line 896: this concept should be expanded to all the agonists of PRRs to include RNA sensing (RNA oncolytic viruses, BO-112), DNA sensing (DNA oncolytic viruses, STING agonists) and TLRs. Figure 4: PRR agonists could be included among therapeutic strategies in “IFNg pathway deficient”

We thank the reviewer for this suggestion and we have now added “oncolytic viruses” to Concluding remarks (section 5, line 922), and also amended Figure 4 to reflect this change.

We are not entirely convinced that use of PRR- and more specifically, TLR agonists will help overcome IFNg pathway deficiency, and/or MHC-I loss in solid tumors. We are fully aware that many relevant topics were not covered in this review (e.g. adoptive cell therapies or vaccination strategies) and while use of PRR agonists (e.g. as adjuvants) has its place in oncoimmunology, we believe these would be better suited to a different review.